# An ESR1-Related Gene Signature Identifies Head and Neck Squamous Cell Carcinoma with Imputed Susceptibility to Endocrine Therapy

**DOI:** 10.3390/ijms25021244

**Published:** 2024-01-19

**Authors:** Firas Almouhanna, Jochen Hess

**Affiliations:** 1Department of Otorhinolaryngology, Head and Neck Surgery, Heidelberg University Hospital, 69120 Heidelberg, Germany; 2Research Group Molecular Mechanisms of Head and Neck Tumors, German Cancer Research Center (DKFZ), 69120 Heidelberg, Germany

**Keywords:** head and neck cancers, estrogen receptor alpha, risk model, endocrine therapy

## Abstract

Head and neck squamous cell carcinoma (HNSCC) is associated with high morbidity and mortality. New personalized treatment strategies represent an unmet medical need to improve the overall survival and the quality of life of patients, which are often limited by the toxicity of established multimodal treatment protocols. Several studies have reported an increased expression of the estrogen receptor 1 (ESR1) in HNSCC, but its potential role in the disease outcome of these tumors remains elusive. Using an integrative analysis of multiomics and clinical data from The Cancer Genome Atlas (TCGA)-HNSC, we established a prognostic risk model based on an ESR1-related 25-gene set. The prognostic value was confirmed in an independent cohort of HNSCC and other solid tumors from TCGA. Finally, we performed in silico drug sensitivity modeling to explore potential vulnerabilities for both risk groups. This approach predicted a higher sensitivity for HNSCC, with prominent ESR1 pathway activity under treatment with specific estrogen receptor modulators. In conclusion, our data confirm the involvement of ESR1-related pathway activity in the progression of a defined subset of HNSCC, provide compelling evidence that these tumors share a specific vulnerability to endocrine therapy, and pave the way for preclinical studies and clinical trials to demonstrate the efficacy of this new therapeutic option.

## 1. Introduction

Head and neck squamous cell carcinoma (HNSCC) originates from the mucosal epithelia of the oral cavity, larynx, and pharynx and is one of the most common cancers, with approximately 900,000 new cases reported worldwide each year [1]. Although HPV-positive HNSCCs are characterized by certain molecular features and a favorable disease outcome compared to those tumors without a history of HPV infection [2], the current standard of care for both subtypes of the disease in locally advanced tumors is high-dose chemoradiotherapy (mainly for oropharyngeal tumors) or surgery followed by chemoradiotherapy (mainly for oral cavity, laryngeal, and hypopharyngeal tumors), whereas early-stage tumors are treated with irradiation or surgery alone [3,4]. The adequate treatment of advanced HNSCC remains a major challenge due to the high risk of treatment failure, which is associated with a dismal prognosis, and high mortality rates, associated with the toxicity of classical treatment regimens [1]. Therefore, a high-quality multidisciplinary team is an essential component to improving clinical outcomes [5], and a molecular stratification of HNSCC subgroups that could benefit from innovative targeted therapy-based regimens is an urgent medical need in order to improve not only the survival but also the quality of life of patients and to guide the decision making process of deintensification of classical treatment, thereby reducing its toxicity.

Recent studies have shown a correlation between estrogen receptor 1 (ESR1) expression and a more favorable survival for oropharyngeal squamous cell carcinoma (OPSCC), suggesting a potential role for estrogen receptor alpha (ERα) signaling in the development and maintenance of these tumors [6]. A recent report identified a group of HPV-positive HNSCC tumors with features of viral integration and high levels of ESR1 expression [7]. However, previous reports referred to ESR1 levels at either the mRNA or protein expression. It is worth noting that multiple factors contribute to the biological activities of ESR1 rather than its mere expression. Therefore, the relationship between the activity of ERα and the survival in HNSCC remains to be investigated.

In this study, we aimed to identify subgroups of HNSCC based on an ESR1-related gene set and to investigate clinical variables in these subgroups. Furthermore, the main objective of this study was to establish an ESR1-related prognostic risk model and to identify potential vulnerabilities for pharmacologic intervention in molecularly defined subgroups of patients.

## 2. Results

### 2.1. ESR1-Related Gene Set in the TCGA-HNSC Cohort

The main objective of this study was to identify an ESR1-related gene set and to elucidate its impact on the pathogenesis and prognosis of HNSCC patients. To achieve this goal, tumors of the oral cavity, larynx, and pharynx from The Cancer Genome Atlas Head and Neck Squamous Cell Carcinoma (TCGA-HNSC) cohort (n = 500) were ranked based on their ESR1 transcript levels and classified into ESR1-High (top quartile), ESR1-Low (bottom quartile), and ESR1-Moderate (all others) (Appendix A). Significant differences in ESR1 expression between the ESR1-High and ESR1-Low subgroups were confirmed at both the transcript and protein levels (Appendix A), and a differential gene expression analysis between the two subgroups was performed using the Limma voom and EdgeR packages in R. The analysis revealed 192 and 775 differentially expressed genes (DEGs) with a |log2FC| > 1.5 and adj. *p*-value < 0.05, respectively (Appendix A). Overall, an ESR1-related gene set of 139 common DEGs (Appendix A) was identified and used for further analysis.

### 2.2. Stratification of HNSCC Subgroups Based on the ESR1-Related 139-Gene Set

The unsupervised hierarchical clustering of the TCGA-HNSC cohort based on the transcript levels of the ESR1-related 139-gene set identified two main clusters, each divided into two subclusters: A1, A2, B1, and B2 (Figure 1A). The cross-tabulation analysis confirmed a significant enrichment of HPV16-positive OPSCC in males in subcluster B2, with the highest ESR1 transcript and protein levels, whereas subcluster A1, with the lowest ESR1 transcript levels and a high expression of the down-regulated DEGs in the ESR1-related 139-gene set, was enriched for HPV-negative laryngeal tumors (Appendix A and Figure 1B). As expected, patients in subcluster B2 had the best 5-year overall survival (OS) in a Kaplan Meier analysis, whereas patients in subcluster B1 had the worst prognosis (Figure 1C). As both subclusters were characterized by a similar expression pattern of up- and down-regulated DEGs from the ESR1-related 139-gene set, this finding indicates a context-dependent impact of the ERα pathway activity and its related genes on the disease outcome.

### 2.3. Establishment of a Prognostic Risk Model with TCGA-HNSC as a Training Cohort

A Least Absolute Shrinkage and Selection Operator (LASSO)-penalized Cox regression analysis was performed to prioritize the most clinically relevant candidate genes from the ESR1-related 139-gene set, using the 5-year overall survival (OS) as the clinical endpoint, and to establish a prognostic risk model. This analysis revealed an ESR1-related 25-gene set, which was used to calculate a prognostic risk score for all cases in the TCGA-HNSC cohort (Appendix A). A Lambda analysis was used to define the best cut-off separating cases of the TCGA-HNSC cohort into high-risk and low-risk groups (Figure 2A and Appendix A). The prognostic value of the newly established risk model was confirmed by Kaplan–Meier analysis considering 5-year progression-free intervals (PFI), disease-specific survival (DSS), and OS (Appendix A), as well as univariate and multivariate Cox regression hazard models (Appendix A). Moreover, a subgroup analysis confirmed a significant performance of the risk model in all subgroups tested, with the highest hazard ratio in the subgroup of HPV16-positive tumors (Figure 2B). As expected, HPV16-positive OPSCC were highly enriched in the low-risk group compared to the high-risk group. In addition, patients in the low-risk group were significantly younger, whereas perineural invasion was more frequent in the high-risk group (Figure 2A and Appendix A). Finally, ESR1 expression at transcript and protein levels and gene set variation analysis (GSVA) scores for the PID_ERA_GENOMIC_PATHWAY gene set were evaluated in both risk groups and were significantly higher in the low-risk group compared to the high-risk group (Figure 2C).

### 2.4. Validation of the Risk Model in an Independent HNSCC Cohort and Other Solid Tumors from TCGA

To confirm the prognostic performance of the risk model in tumors of the oral cavity, larynx, and pharynx from an independent HNSCC cohort, bulk RNA-seq data from Clinical Proteomic Tumor Analysis Consortium (CPTAC)-HNSC (n = 104) were used to calculate risk scores and to divide all patients into high-risk and low-risk groups. Again, the low-risk group showed a significantly better 5-year OS and higher GSVA scores for the PID_ERA_GENOMIC_PATHWAY gene set, indicating higher ESR1-related pathway activity (Figure 3A,B). However, it is worth noting that no significant difference in ESR1 transcript levels was found between the two risk groups, confirming our basic assumption that the ESR1 transcript levels are not necessarily an adequate indicator for pathway activity (Figure 3B). The prognostic value of the risk model was also analyzed beyond HNSCC by calculating the risk scores for other solid tumors from TCGA, namely, prostate adenocarcinoma (TCGA-PRAD), pancreatic adenocarcinoma (TCGA-PAAD), esophageal carcinoma (TCGA-ESCA), liver hepatocellular carcinoma (TCGA-LIHC), glioblastoma (TCGA-GBM), bladder urothelial carcinoma (TCGA-BLCA), lung adenocarcinoma (TCGA-LUAD), ovarian serous cystadenocarcinoma (TCGA-OV), breast invasive carcinoma (TCGA-BRCA), colorectal adenocarcinoma (TCGA-COAD), lung squamous cell carcinoma (TCGA-LUSC), uterine corpus endometrial carcinoma (TCGA-UCEC), kidney renal clear cell carcinoma (TCGA-KIRC), and cervical squamous cell carcinoma (TCGA-CSCC). The hazard ratios showed a similar trend to HNSCC in 9 out of 14 cohorts, with a significant difference between low-risk and high-risk tumors in 7 cohorts (Figure 3C). A Kaplan–Meier for the nine cohorts following the HNSCC trend (combined) revealed a significantly better 5-year OS for the low-risk group (Figure 3D).

### 2.5. In Silico Drug Response Prediction

The ultimate goal of this study was to identify specific vulnerabilities as potential drug targets for a more effective and/or less toxic treatment of the patients stratified by the newly established prognostic risk model. As previous data indicated a higher ESR1-related pathway activity in the low-risk group, we particularly focused our attention on well-established modulators and degraders of ERα, namely, Tamoxifen and Fulvestrant. RNA-seq and drug screening data for human cancer cell lines from the Genomics and Drug Sensitivity in Cancer (GDSC) project were used to train a ridge regression model for drug response with the calcPhenotype function of the oncoPredict package in R, which was fitted to the gene expression data from the TCGA-HNSC cohort. Indeed, the imputed drug sensitivity scores of the trained model were significantly lower for tumors in the low-risk group compared to the high-risk group, indicating a better drug response output with fulvestrant or tamoxifen (Figure 4A). The significantly higher sensitivity of low-risk tumors was confirmed, at least in part, for CPTAC-HNSC as an independent HNSCC cohort, as well as for selected other solid tumors from TCGA (PRAD, PAAD, ESCA, LIHC, GBM, BLCA, LUAD, OV, BRCA), with similar performance of the prognostic risk model (Figure 4B,C).

## 3. Discussion

Numerous experimental and clinical studies have reported an important role for estrogen signaling in the development and progression of various cancers [8]. Primarily, ESR1 expression is associated with various pathological aspects of breast cancer; however, it is worth noting that ESR1 positivity is associated with a better prognosis in these patients and could be targeted by specific estrogen receptor modulators [9]. Furthermore, a link between estrogen signaling and HPV infection has been established in the development of cervical cancer [10]. The exposure of cervical cancer cells to estrogen increases the expression of two major HPV16 and HPV18 oncogenes, namely, E6 and E7, suggesting a role for both players in cervical carcinogenesis [11]. HPV16 infection is a well-established risk factor for HNSCC, and HPV-positive HNSCCs have a much better prognosis than their HPV-negative counterparts [10,12]. Recent studies have reported a prominent ESR1 expression in smaller tumors, particularly enriched in HPV-positive OPSCC, and its association with a better prognosis in OPSCC, even independently of the HPV status, and suggested the addition of endocrine therapy for a safe dose reduction in cytotoxic treatment, particularly in HPV-positive OPSCC, or as an alternative systemic treatment that could address metastatic tumors [6]. This notion is supported by several recent studies providing experimental evidence that tamoxifen or fulvestrant treatment of HNSCC cell lines increases apoptosis [13], reduces invasion in combination with EGFR inhibition [13], and sensitizes tumor cells to fractionated irradiation [14]. These data do not only suggest a functional interplay between estrogen signaling and HPV infection in the pathogenesis of OPSCC, but also indicate a prognostic value of ESR1 signaling in both HPV-positive and HPV-negative HNSCCs. However, the underlying cellular and molecular principles are elusive and remain to be described. Of particular interest are more robust markers that reliably indicate active ERα signaling and help identify HNSCC patients as potential candidates for well-established therapies targeting ERα. Our data confirm previous findings showing a strong correlation between ESR1 transcript levels, oropharynx as the predominant anatomical subsite, HPV status, and lower pathological grading. Interestingly, the stratification of HNSCC based on the expression of the ESR1-related 139-gene set revealed distinct clusters with variable ESR1 transcript and protein levels and clinical outcomes. These data strongly support the notion that the mere detection of an ESR1 transcript or protein expression levels is not sufficient to predict the disease outcome in all tumor samples from different subsites and with different molecular characteristics, especially those that are related to the HPV status, and this obviously represents a limitation of previous studies.

Based on the RNA expression of the prognostic 25-gene set identified in this study, we stratified HNSCC patients into different risk groups with significant differences in the 5-year OS, and these differences were significant even after adjustment for other relevant clinical variables. It is also worth noting that this finding was independent of the ESR1 expression in HPV-negative HNSCCs in the training and validation cohorts, a finding that confirms the robust prognostic value of our gene set and overcomes the aforementioned limitations of other studies. Furthermore, higher GSVA scores for the “PID_ERA_GENOMIC_PATHWAY” gene set in the low-risk group of the independent cohorts indicate a higher activity of ERα in these tumors. Interestingly, another study [6] also reported an increased enrichment of the “PID_ERA_GENOMIC_PATHWAY” that is associated with higher ESR1 expression in OPSCCs, which is consistent with our finding in the training cohort. However, our data show that ESR1 expression is not associated with the high “PID_ERA_GENOMIC_PATHWAY” GSVA scores in HPV-negative tumors of the validation cohort, a phenomenon that may be partially explained by the contextual role of ERα and its complex interactions with numerous transcription factors in different tissues.

The need to identify robust markers to predict the response to endocrine therapy in ESR1-related tumors, such as breast cancer, has received increasing attention, because only 50–70% of patients with receptor-positive tumors (identified by high ESR1 expression) respond to this treatment. This phenomenon demonstrates the limited predictive value of ESR1 expression [15]. In our study, tumors of the low-risk group showed a higher imputed sensitivity to a modulator (tamoxifen) and a degrader (fulvestrant) of ERα in two independent HNSCC cohorts, and in the combined cohort of nine other solid tumors from TCGA. This finding demonstrates that the newly established risk model not only serves as a reliable prognosticator, but also has a strong potential as a therapeutic classifier to identify patients who may benefit from an endocrine therapy targeting ERα. A lower toxicity of endocrine therapy could reduce the high morbidity that is associated with the acute and chronic treatment-related toxicity of classical regimens. However, a more comprehensive analysis of the mutational landscape, signaling pathways, and gene regulatory networks in individual risk groups is needed to further elucidate underlying molecular features and to identify potential mechanisms of resistance to endocrine treatment in HNSCC patients.

Our study is not without limitations. Bulk RNA-seq data could be biased by ESR1 expression and/or pathway activity in non-malignant cells of the TME. To address this issue, single-cell RNA-seq data from HNSCC samples of recently published studies could be analyzed in order to assess the association between ESR1 expression and the ESR1-related 25-gene set in malignant epithelial cells versus non-malignant cells and to confirm the potential vulnerability of cancer cells to endocrine treatment at the single-cell level. In addition, the retrospective study design including bioinformatics algorithms and in silico data analysis warrants further validation in prospective clinical trials and an experimental proof-of-concept in appropriate preclinical models.

In conclusion, our data further substantiate the clinical relevance of ESR1-related pathway activity in the carcinogenesis of a subset of HNSCC, in particular HPV16-positive OPSCC, and provide a reliable prognostic risk model based on a newly established ESR1-related 25-gene set. Molecular stratification using this prognostic risk model may identify HNSC patients who might benefit from endocrine treatment in order to improve the disease outcome and/or quality of life. 

## 4. Materials and Methods

### 4.1. Cohorts

RNA-seq data (FPKM) for the HNSC training cohort (n = 500) and other solid tumors from TCGA (TCGA-PRAD (n = 491), TCGA-PAAD (n = 176), TCGA-ESCA (n = 163), TCGA-LIHC (n = 369), TCGA-GBM (n = 159), TCGA-BLCA (n = 404), TCGA-LUAD (n = 504), TCGA-OV (n = 374), TCGA-BRCA (n = 1089), TCGA-COAD (n = 453), TCGA-LUSC (n = 494), TCGA-UCEC (n = 542), TCGA-KIRC (n = 529), and TCGA-CSCC (n = 303)) were downloaded (accessed on 10 October 2020) from GDC data portal using the Subio platform available at https://www.subioplatform.com. Protein expression (RPPA), clinical, and survival data for TCGA cohorts were downloaded from cBioPortal (TCGA, Firehose Legacy) at https://www.cbioportal.org/study/summary?id=hnsc_tcga (accessed on 13 November 2020). RNA-seq (RSEM) and clinical data for the CPTAC-HNSC cohort were downloaded from LinkedOmics portal at http://linkedomics.org/login.php (accessed on 5 August 2021).

### 4.2. Differential Gene Expression

Tumors from the TCGA-HNSC training cohort (n = 500) were classified into three groups according to their ESR1 transcript levels: ESR1-High (representing the top 25%), ESR1-Low (representing the bottom 25%), and ESR1-Moderate (representing all other tumors). A two-tailed t-test was used to calculate *p* values and to detect significant differences in expression among tumors of both groups in the TCGA-HNSC cohort. Differentially expressed genes (DEGs) between the ESR1-High vs. ESR1-Low groups were identified using Limma-voom [16] and edgeR [17] packages in R studio (3.6.0). The ESR1-related gene set (n = 139 genes) representing only common DEGs (isolated by the two R packages used) was defined.

### 4.3. Unsupervised Hierarchical Clustering

Unsupervised hierarchical clustering was performed using the ComplexHeatmap package [18] in R studio (3.6.0), applying the following settings: clustering for rows “no clustering”, clustering distance for columns “Euclidean”, clustering method for columns “ward.D2”.

### 4.4. Cross-Tabulation Analysis

Cross-tabulation analysis was performed with the chisq.test function in R studio (3.6.0), and *p* values were calculated with the Chi square test.

### 4.5. GSVA Analysis

Gene set variation analysis (GSVA) was used to calculate the enrichment scores for the selected gene set “PID_ERA_GENOMIC_PATHWAY”, downloaded from the MsigDatabase https://www.gseamsigdb.org/gsea/msigdb/ (accessed on 1 November 2023) using the GSVA package [19] in R studio (3.6.0). Significant differences in GSVA scores between groups were tested with a two-tailed *t*-test.

### 4.6. Survival Analysis

Kaplan–Meier plots were conducted to investigate differences in survival among patient groups, and log-rank tests were applied to calculate *p* values using survminer [20] and survival [21] packages in R studio (3.6.0). Univariate and multivariate Cox proportional hazards regression models were conducted with either the IBM SPSS Statistics software (version 25) (IBM Deutschland, Ehningen, Germany) or the survival package in R studio (3.6.0).

### 4.7. Risk Model

LASSO-penalized Cox regression analysis was performed to prioritize most relevant prognostic DEGs for 5-year overall survival using the glmnet package [22] in R studio (3.6.0). The risk score for each tumor was calculated using glmnet package (s = lambda.min and type = “response”) based on the following coefficient values for the selected 25 genes: ZNF831 = −0.4258735, FCRL3 = −0.1400017, SOX30 = −0.0956562, LHX9 = −0.0291356, SLC13A5 = −0.0248776, NOBOX = −0.0224077, PRAP1 = −0.0195581, CYP1A1 = −0.0165553, ATP13A5 = −0.012558, NEUROD2 = −0.0118979, WDFY4 = −0.0061702, CYP4X1 = −0.00601, FOXH1 = −0.0059322, SFRP1 = −0.0037341, LIM2 = −0.0013229, FAM25A = −0.0010163, CALB1 = 0.00163639, CHRDL1 = 0.00211768, LGI3 = 0.00365953, CCL26 = 0.00498837, OLFM4 = 0.0054605, GAST = 0.00735744, SMYD1 = 0.00836536, HRH2 = 0.03578589, and HOXB8 = 0.08355879. The Maxstat package [23] in R studio (3.6.0) was used to set the best cut-off in order to classify cases from TCGA-HNSC into high-risk and low-risk groups, representing unfavorable and favorable overall survival, respectively.

Transcriptome data were used to calculate risk scores for patients of the independent HNSCC validation cohorts and other solid tumors from TCGA. The coefficient of each risk gene was multiplied by its corresponding transcriptome value; the mathematical addition of the outcome values for each patient represents the patient’s risk score. Maxstat package in R studio (3.6.0) was used again to define the best cut-off and to separate the patients into high-risk and low-risk groups.

### 4.8. In Silico Drug Response Prediction

The calcPhenotype function of the oncoPredict package [24] was used in R on large-scale gene expression (rma normalized) and drug response data from the GDSC2 library that were prepackaged in the used package. Parameters of the calcPhenotype function were determined as described in the original vignette. The built ridge regression model was applied on the RNA-seq data of the TCGA-HNSC training and CPTAC-HNSC validation cohorts separately, and on other solid tumor cohorts from TCGA, which were combined to predict response to the selected drugs. Significant differences among the risk groups were calculated with a two-tailed Student’s *t*-test.

### 4.9. Data Visualization

Graphics (point plots, violin plots, volcano plots) were generated using the ggplot2 package [25] in R studio (3.6.0). Venn diagrams were generated using Venny 2.1.0- BioinfoGP [22]. Heatmaps were generated using the ComplexHeatmap package [18] in R studio (3.6.0). Kaplan–Meier plots and the numbers at risk tables were generated using the survminer package [20] in R studio (3.6.0). Forest plots and bar plots were generated using Microsoft Excel. Graphics were edited by Inkscape 0.92 software (Free Software Foundation, Inc. Boston, MA, USA).

## Figures and Tables

**Figure 1 ijms-25-01244-f001:**
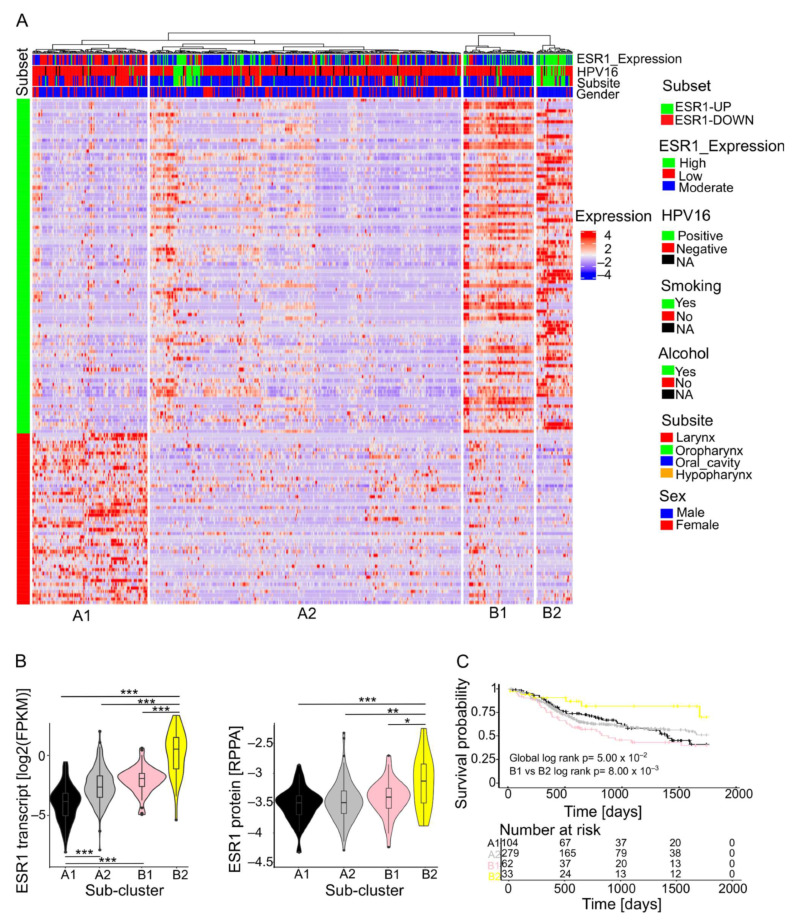
HNSCC subgroups based on the ESR1-related 139-gene set. (**A**) Heat map shows an unsupervised hierarchical clustering of cases from TCGA-HNSC based on transcript levels of the ESR1-related 139-gene set and their association with indicated risk factors and clinical variables; (**B**) violin plots show ESR1 transcript (left) or protein levels (right) for individual subclusters; (**C**) Kaplan–Meier plot showing the 5-year overall survival for each subcluster. Numbers of patients at risk at the indicated time points are given below the graph. * *p* ≤ 0.05, ** *p* < 0.005, *** *p* < 0.0005.

**Figure 2 ijms-25-01244-f002:**
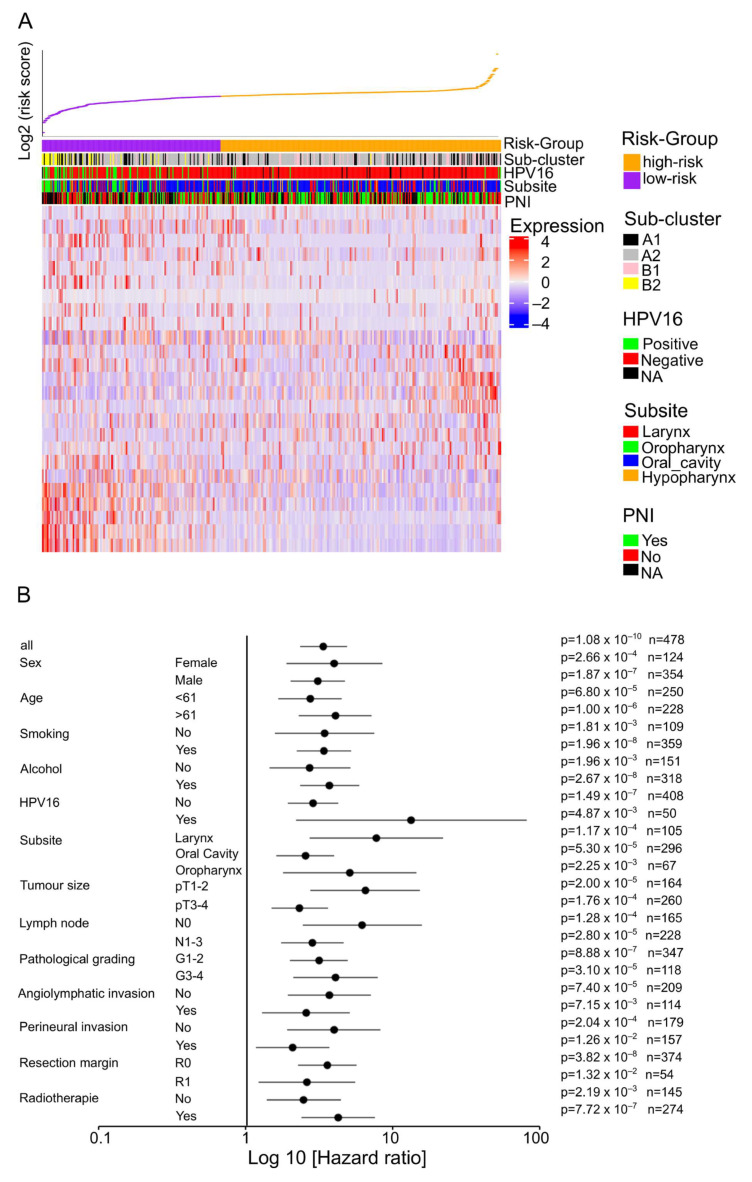
Establishment of a prognostic risk model based on the ESR1-related 25-gene set for TCGA-HNSC. (**A**) Heat map illustrates transcript levels of the prognostic ESR1-related 25-gene set with ranked columns for cases according to the risk score. (**B**) Forest plot shows hazard ratios (HRs) and 95% confidence intervals (95% CIs) for 5-year overall survival of the indicated features and variables of TCGA-HNSC with the low-risk group as a reference. (**C**) Violin plots demonstrate significantly higher ESR1 transcript (upper left) and protein levels (upper right) for the low-risk group, as well as significantly higher GSVA scores for the PID_ERA_GENOMIC_PATHWAY gene set (lower) for tumors of the low-risk group. * = *p* ≤ 0.05, *** = *p* < 0.0005.

**Figure 3 ijms-25-01244-f003:**
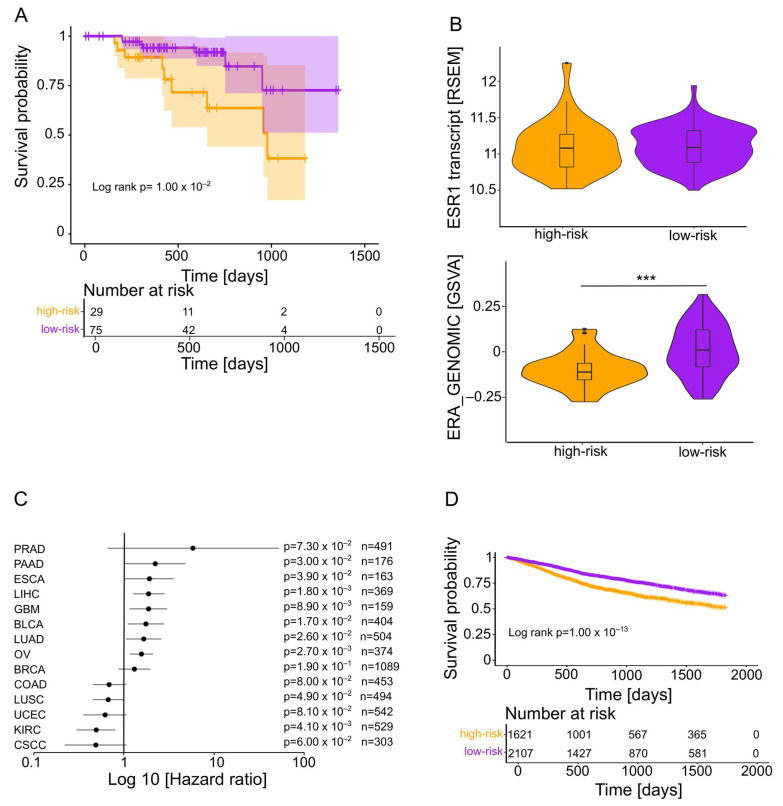
Validation of the prognostic risk model in CPTAC-HNSC and other cohorts from TCGA. (**A**) Kaplan–Meier plot shows a favorable 5-year OS for the low-risk group compared to the high-risk group. Numbers of patients at risk at the indicated time points are given below the graph. (**B**) Violin plots demonstrate no significant difference in ESR1 transcript levels between the two risk groups (top) but a significantly higher GSVA score for the PID_ERA_GENOMIC_PATHWAY gene set in the low-risk group (bottom). (**C**) Forest plot shows hazard ratios (HRs) and 95% confidence intervals (95% CIs) for 5-year overall survival with the low-risk group as reference for indicated cohorts from TCGA. (**D**) Kaplan–Meier plot shows a significant difference in 5-year overall survival between the low-risk group and high-risk for 9 selected TCGA cohorts (combined). Numbers of patients at risk at the indicated time points are given below the graph. *** *p* < 0.0005.

**Figure 4 ijms-25-01244-f004:**
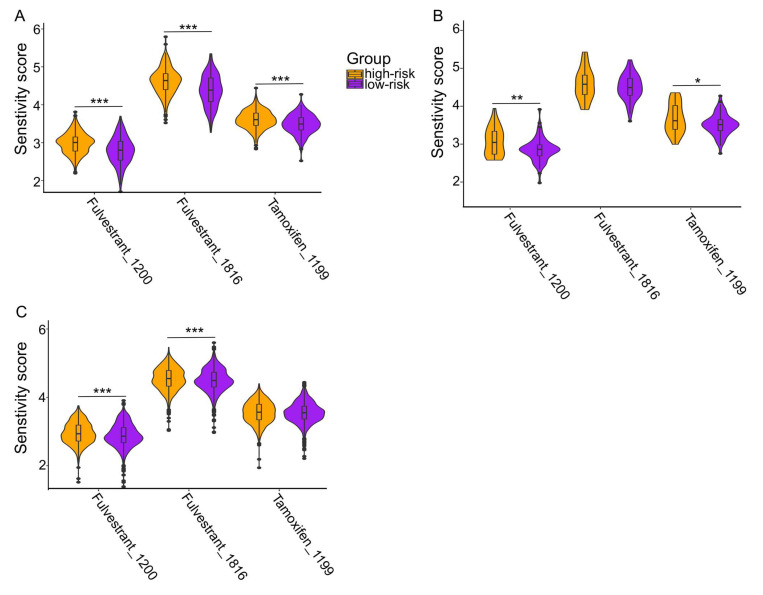
Imputed drug response for an ERα modulator and degrader. Violin plots show lower drug sensitivity scores of fulvestrant and tamoxifen for tumors of low-risk group patients in TCGA-HNSC (**A**), CPTAC-HNSC (**B**), and the combined cohort of other solid tumors from TCGA (**C**). * *p* ≤ 0.05, ** *p* < 0.005, *** *p* < 0.0005.

## Data Availability

The data generated in this study are available within the article and its Appendix A.

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
