# Peer review of "An ESR1-Related Gene Signature Identifies Head and Neck Squamous Cell Carcinoma with Imputed Susceptibility to Endocrine Therapy"

_ijms, 2024, doi:10.3390/ijms25021244_

Round 1

Reviewer 1 Report

Comments and Suggestions for Authors

The topic is important and timely. The text reads well in general. I have minor suggestions. In the introduction, treatment strategies and the multidisciplinary team management should be stressed. For instance, chemoradiotherapy is the standard of care in locally advanced oropharyngeal cancer (line 34) and a high-quality multidisciplinary team is an essential component to improve clinical outcomes (line 40-41). Therefore, text would be enhanced by addition of references, such as PMID: 33239190 and PMID: 30426243 to better contextualize the issue at hand in oncologic scenario. Limits and strengths of the study supporting evidence and validity of the research should be better reported.

Author Response

Thank you very much for taking the time to review this manuscript. Please find the detailed responses below (responses are highlighted in red below their corresponding comments).

Comment1: The topic is important and timely. The text reads well in general. I have minor suggestions. In the introduction, treatment strategies and the multidisciplinary team management should be stressed. For instance, chemoradiotherapy is the standard of care in locally advanced oropharyngeal cancer (line 34) and a high-quality multidisciplinary team is an essential component to improve clinical outcomes (line 40-41). Therefore, text would be enhanced by addition of references, such as PMID: 33239190 and PMID: 30426243 to better contextualize the issue at hand in oncologic scenario.

Response1: As proposed, the information and required references have been added to the Introduction of the revised manuscript on page 1, lines 34-37 and lines 40-41.

Comment2: Limits and strengths of the study supporting evidence and validity of the research should be better reported.

Response2: As suggested, the limitations and conclusions of the study have been described in the revised manuscript on pages 10-11, lines 247-261.

Reviewer 2 Report

Comments and Suggestions for Authors

Brief summary

The paper focuses the attention on an interesting topic.

This is a clear paper, with a good structure and fluent.

The conclusions are interesting and the results are clear

No ethical problems are found in this study.

However, some criticisms are present in the paper.

General concepts

Please specify that TGCA-HNSC refers to “The Cancer Genome Atlas Head-Neck Squamous Cell Carcinoma” at least the first time you cited it

Please specigy what hystotypes are the squamous cell carcinoma you analyse if possibile

Please specify at least once the sequent abbreviations:

TCGA-PRAD (n=491), TCGA-PAAD (n=176), TCGA-ESCA (n=163), 232 TCGA-LIHC (n=369), TCGA-GBM (n=159), TCGA-BLCA (n=404), TCGA-LUAD (n=504), 233 TCGA-OV (n=374), TCGA-BRCA (n=1089), TCGA-COAD (n=453), TCGA-LUSC (n=494), 234 TCGA-UCEC (n=542), TCGA-KIRC (n=529), TCGA-CSCC (n=303)

Author Response

Thank you very much for taking the time to review this manuscript. Please find the detailed responses below (responses are highlighted in red below their corresponding comments).

Comment1: The paper focuses the attention on an interesting topic. This is a clear paper, with a good structure and fluent. The conclusions are interesting and the results are clear. No ethical problems are found in this study. However, some criticisms are present in the paper. General concepts: Please specify that TGCA-HNSC refers to “The Cancer Genome Atlas Head-Neck Squamous Cell Carcinoma” at least the first time you cited it.

Response1: Has been done on page 2, lines 64-65.

Comment2: Please specify what hystotypes are the squamous cell carcinoma you analyse if possible

 Response2:  The TCGA and CPTAC HNSCC analyzed in this study originate from the mucosal epithelia of the oral cavity, larynx, and pharynx. This information has been included in the revised manuscript on page 1 lines 29-30, page 2 lines 64-65, page 6 lines 127-128. In addition, squamous cell carcinoma of the esophagus (TCGA-ESCA), bladder (TCGA-BLCA), lung (TCGA-LUSC), and cervix (TCGA-CSCC) were analyzed. However, a more detailed specification of the histotype for these tumors is not provided in the publicly available data deposited in the GDC data portal or in the original publications.

Comment3: Please specify at least once the sequent abbreviations:

TCGA-PRAD (n=491), TCGA-PAAD (n=176), TCGA-ESCA (n=163), 232 TCGA-LIHC (n=369), TCGA-GBM (n=159), TCGA-BLCA (n=404), TCGA-LUAD (n=504), 233 TCGA-OV (n=374), TCGA-BRCA (n=1089), TCGA-COAD (n=453), TCGA-LUSC (n=494), 234 TCGA-UCEC (n=542), TCGA-KIRC (n=529), TCGA-CSCC (n=303)

Response3:  Has been done on pages 6-7, lines 137-144.

Reviewer 3 Report

Comments and Suggestions for Authors

Would it be possible explain HNSCC homogeneity and heterogeneity in HPV tumors related to prognosis, and it is possible do a hypothesis in which established the association with ESR1 and posible therapies with tamoxifen and fulvestrant?

Line 220 to 224 The paragraph it is interesting, however it seems incomplete, authors describe the effective treatment against ER-alpha (tamoxifen-fulvestrant) it is seems effective in low grade tumors and HPV positive. Would it be possible establish a better description about it and compare with high-grade, heterogeneus and HPV negative tumors?

It is important describe conclusion of the study made it. explaining: advantages and limitants and futures studies related with this interesting topic. 

Author Response

Thank you very much for taking the time to review this manuscript. Please find the detailed responses below (responses are highlighted in red below their corresponding comments). Please see the additional figure in the attachment.

Comment1:  Would it be possible explain HNSCC homogeneity and heterogeneity in HPV tumors related to prognosis, and it is possible do a hypothesis in which established the association with ESR1 and posible therapies with tamoxifen and fulvestrant?

Response1:  In the submitted manuscript, we have already shown by multivariate Cox regression hazard models that the newly established risk model based on the ESR1-related 25-gene set is an independent prognostic factor after adjustment for other known risk factors, including HPV16 (Supplementary Table S4). Although HPV16-positive OPSCC were highly enriched in the low-risk group as expected (Supplementary Table S5), it is worth noting that the subgroup analysis showed the highest hazard ratio in the subgroup of HPV16-positive tumors (Figure 2B). Accordingly, we do not believe that tumor heterogeneity based on cellular or molecular differences between HPV-negative and HPV-positive HNSCC has a strong impact on the prognostic performance of the newly established risk model. This assumption is further supported by the validation of the risk model in an independent HPV-negative HNSCC cohort (CPTAC-HNSC) and other solid tumors from TCGA (Figure 3A, Figure 3C-D).

As suggested by Koenigs et al. (PMID: 30715409), the addition of endocrine therapy has clinical potential for a safe dose reduction of cytotoxic treatment, particularly in HPV-positive OPSCC, or as an alternative systemic treatment that could target metastatic tumors. This hypothesis is supported by several recent studies providing experimental evidence that tamoxifen or fulvestrant treatment of HNSCC cell lines increases apoptosis (PMID: 17355262), reduces invasion in combination with EGFR inhibition (PMID: 19825947), and sensitizes tumor cells to fractionated irradiation (PMID: 28166815). The discussion of this hypothesis and relevant references have been included in the revised manuscript on pages 9-10 lines 195-201.

Comment2: Line 220 to 224 The paragraph it is interesting, however it seems incomplete, authors describe the effective treatment against ER-alpha (tamoxifen-fulvestrant) it is seems effective in low grade tumors and HPV positive. Would it be possible establish a better description about it and compare with high-grade, heterogeneus and HPV negative tumors?

Response2: We agree that the data presented indicate that patients with HPV16-positive OPSCC and a low-risk phenotype according to the risk model based on the ESR1-related 25-gene set are likely to benefit from endocrine treatment with either tamoxifen or fulvestrant. However, the inferred drug sensitivity scores for fulvestrant and tamoxifen are also lower for low-risk tumors from CPTAC-HNSC, which represents an HPV-negative HNSCC cohort (Figure 4B). In line with this finding, we analyzed the drug sensitivity scores of tamoxifen and fulvestrant for the subset of HPV-negative tumors from TCGA-HNSC and confirmed a higher sensitivity for low-risk compared to high-risk tumors stratified by the new risk model (Figure for reviewer). In conclusion, these data suggest a clinical potential for endocrine therapy in both HPV-positive and HPV-negative HNSCC stratified by the prognostic risk model. Ongoing studies in preclinical models aim to provide the experimental proof-of-concept, but are beyond the scope of this manuscript.

Comment3: It is important describe conclusion of the study made it. explaining: advantages and limitants and futures studies related with this interesting topic. 

Response3: As suggested limitations, conclusions and future directions have been described in the revised manuscript on pages 10-11, lines 247-261.

Round 2

Reviewer 1 Report

Comments and Suggestions for Authors

Revision ok. But ref [5] is an inappropriate citation in line 40-41. Please verify reference. 

Author Response

Thank you very much for your revision.

Comment: Revision ok. But ref [5] is an inappropriate citation in line 40-41. Please verify reference. 

Response: Reference [5] has been verified.